# The Relationship between Bone Conduction Hearing Threshold Shifts after Surgery for Chronic Otitis Media with Cholesteatoma According to STAM, EAONO/JOS, and SAMEO-ATO Classifications

**DOI:** 10.3390/jcm11154481

**Published:** 2022-08-01

**Authors:** Jan Mejzlik, Viktor Chrobok, Michal Homolac, Tomas Valenta, Anna Svejdova, Michal Cerny, Maja Striteska, Jana Krtickova, Lukas Skoloudik

**Affiliations:** 1Department of Otorhinolaryngology and Head and Neck Surgery, University Hospital Hradec Kralove, 500 05 Hradec Kralove, Czech Republic; chrobok@fnhk.cz (V.C.); michal.homolac@fnhk.cz (M.H.); tomas.valenta@fnhk.cz (T.V.); michal.cerny@lfhk.cuni.cz (M.C.); maja.striteska@fnhk.cz (M.S.); jana.krtickova@fnhk.cz (J.K.); lukas.skoloudik@fnhk.cz (L.S.); 2Faculty of Medicine in Hradec Kralove, Charles University, 500 03 Hradec Kralove, Czech Republic

**Keywords:** bone conduction, classification of cholesteatoma, mastoidectomy, chronic otitis media, SAMEO-ATO, STAM

## Abstract

Background: This study focuses on the hearing threshold for bone conduction (BC) after middle-ear surgery. Methods: A total of 92 patients (120 ears) were treated for newly diagnosed chronic otitis media with cholesteatoma (2013–2018). BC was examined at frequencies of 0.5, 1, 2, and 4 kHz prior to and 1 year after surgery. STAM classification for cholesteatoma location, EAONO/JOS for stage, and surgery according to SAMEO-ATO classification were applied. The bone conduction threshold was compared for individual frequencies in patients with occurrence/absence of cholesteatoma in different locations. Results: For the occurrence of cholesteatoma in the attic (A), a statistically significant difference was found at 4 kHz (*p* < 0.001), in the supratubal recess (S1) at 4 kHz (*p* = 0.003), and for the mastoid (M) at 0.5 kHz (*p* = 0.024), at 1 kHz (*p* = 0.032), and at 2 kHz (*p* = 0.039). Conclusions: Cholesteatoma location can influence the post-operative hearing threshold for bone conduction.

## 1. Introduction

Although chronic otitis media with cholesteatoma has been known for centuries, its terminology and classification have been developing over time [1]. Due to its locally destructive potential, classifications similar to those of tumor processes have even been proposed [2]. Classification based on the size of cholesteatoma, condition of the ossicles, and the presence of complications aimed to set up an ascending scale which would correspond to clinical importance [3]. Every change in cholesteatoma classification is a challenge for re-examination of complications, hearing gains, or surgical procedures [3,4]. This report evaluates newly originated cholesteatoma STAM, EAONO/JOS, and SAMEO-ATO classifications in relation to post-surgically changed bone conduction. Most patients with chronic otitis media with cholesteatoma develop a decrease in hearing threshold for both air and bone conductions. Changes in bone conduction are usually explained in two ways: chemically by the effect of released toxins into the inner ear perilymph, which impairs the movement and metabolism of inner ear hair cells; and mechanically, by an oval and round window membrane blockade [5]. Serious sensorineural hearing loss can be associated with cochlear fistula [6]. The precise removal of cholesteatoma and maximal preservation of the bone conduction hearing threshold are the primary goals of surgery. Even though the fact that cholesteatoma affects bone conduction is not new, authors are trying to support this information by hard data and statistical analysis. Despite all medical efforts, the hearing threshold of bone conduction continues to deteriorate in some cases [7]. Significant differences were found in the sensorineural hearing loss frequencies at the different cholesteatoma locations [8]. Nevertheless, improvement of the hearing threshold of bone conduction may sporadically even occur [9].

Several factors exert an influence on the post-surgical hearing status. Serious deterioration of the hearing threshold of bone conduction, which Tos describes in ca 1.2% of cases, is influenced by acoustic trauma caused by the drilling, perilymphatic fistula, and undesirable manipulations on middle-ear ossicles [5]. Manipulation of the ossicles can cause a temporary auditory threshold shift at 2 and 4 kHz, but drilling in the mastoid can cause a permanent auditory threshold shift at 4 kHz [10].

The authors analyze whether the location of cholesteatoma according to STAM, stage according to EAONO/JOS, or the type of surgical treatment of chronic otitis media according to SAMEO-ATO classifications can influence the hearing threshold of bone conduction [11,12]. Precise cholesteatoma removal and ossicular chain reconstruction are basic prerequisites for bone conduction improvement. The authors also analyze whether tympanoplasties, primarily designed for air conduction improvement, can also influence bone conduction.

Even though the fact that cholesteatoma affects bone conduction is not new, authors try to support this information by hard data and statistical analysis.

## 2. Materials and Methods

### 2.1. Participants

In the period 2013–2018, altogether 92 patients (120 ears) were surgically treated for newly diagnosed chronic otitis media with cholesteatoma at the Department of Otorhinolaryngology and Head and Neck Surgery, University Hospital Hradec Kralove. The average age was 31.5 years (SD 19.88): 59 males with average age of 31.40 years (SD 2.73) and 33 females with average age of 31.78 years (SD 18.85). The diagnosis of cholesteatoma was confirmed histologically in all cases. High-resolution computed tomography was always used before surgery to evaluate temporal bone erosions.

Inclusion criteria: Chronic otitis media with cholesteatoma (histologically proven), primary surgery, ability of repeated audiological testing, and a one-year follow-up.

Exclusion criteria: Revision surgery, recidivation of cholesteatoma, malignancies in the follow-up period, deaf ear, inability of audiological testing, missing audiological data, and an operation less than one year before.

### 2.2. Audiometry

Pure tone audiometry was performed on affected and non-affected ears at frequencies of 0.5, 1, 2, and 4 kHz, to find the hearing threshold of air and bone conduction, prior to surgery and 1 year after surgery. During the examination, the contralateral ear was masked, in accordance with audiological masking protocols. The examinations were conducted repeatedly to avoid bias. Tympanometry and speech audiometry were conducted before surgery, where applicable.

### 2.3. STAM, Stage of Cholesteatoma, EAONO/JOS and SAMEO-ATO Classifications

Classification of cholesteatoma according to STAM was performed retrospectively, from patients’ files, mainly from the surgical protocols—supratubal recess (S1), tympanic sinus (S2), tympanic cavity (T), attic (A), and mastoid cavity (M)—and EAONO/JOS was employed to decide the stage of cholesteatoma. Stage I cholesteatoma is found at one sublocation, stage II cholesteatoma is found in two or more sublocations, stage III refers to extracranial complications (facial palsy, labyrinthine fistula, labyrinthitis, postauricular abscess or fistula, zygomatic abscess, neck abscess, canal wall destruction, destruction of the tegmen and total adhesion of the pars tensa), and stage IV refers to intracranial complications (purulent meningitis, epidural abscess, subdural abscess, brain abscess, sinus thrombosis, and brain herniation into the mastoid cavity) [11]. Classification of the temporal bone surgery was carried out according to SAMEO classification for mastoid surgery:S1 primary surgery, S2p planned 2nd surgery, S2r revision surgery.A1 endoscopic, A2 transcanal, A3 endaural, A4 retrouaricular approach.Mx no mastoidectomy, M1a canal wall preserved, M1b canal wall preserved and posterior tympanotomy, M2a only scutum removed, M2b scutum removed and posterosuperior canal wall removed, M2c whole canal removed, M3a subtotal petrosectomy with preservation of the otic capsule, M3b subtotal petrosectomy with removal of the otic capsula, including labyrinthectomy.E1 no reconstruction, E2 soft material reconstruction, E2 hard material reconstruction of external ear canal.Ox no reconstruction, O1 partial obliteration, O2 total obliteration of mastoid cavity.

In addition, ATO classification for middle-ear surgery:Ax no bone canal removal, A1 tympanic sulcus widening, A2 external canal widening, A3 total canalplasty.Tx no ear drum grafting, Tn original ear drum preserved, T1 intact membrane reinforcement, T2 partial grafting, T3 total ear drum grafting.Ost tympanic membrane to stapes head, On intact chain preservation, Ox no reconstruction performed, Osd ear drum directly repositioned to stapes head, Oft reconstruction between ear drum and foot plate of the stapes, Ofd ear drum directly repositioned onto stapes footplate [12].

### 2.4. Statistics

The statistical program STATISTICA version: 13.4.0.14, Tibco Software Inc. (Palo Alto, CA, USA), was used for evaluation. Descriptive data are presented as follows: number, mean (standard deviation). The influence of cholesteatoma presence/absence on bone conduction auditory threshold shifts was analyzed. Nonparametric tests, Kruskal–Wallis ANOVA, and the Mann–Whitney U Test were used to evaluate statistical significance. A *p* value of <0.05 was considered significant.

### 2.5. Theory/Calculation

Retrospective analysis of the relationship between cholesteatoma locations and the bone conduction hearing threshold shifts was conducted separately for each frequency of 0.5, 1, 2, and 4 kHz, aiming to find whether changes of bone conduction after cholesteatoma surgery are bound to a certain location of cholesteatoma and whether the possible dependence is frequency specific. Furthermore, a comparison has been made whether the shifts of bone conduction after surgery are influenced by the stage of cholesteatoma according to AEONO/JOS, and the type of surgery according to SAMEO-ATO classifications.

## 3. Results

### 3.1. Changes in Bone Conduction

The mean hearing threshold for air conduction before surgery was 46.88 dB (SD 22.78) at 0.5 kHz, 42.63 dB (SD 25.13) at 1 kHz, 40.00 dB (SD 25.88) at 2 kHz, and 49.26 dB (SD 26.98) at 4 kHz, and after the surgery, respectively, 38.37 dB (SD 23.72), 35.62 dB (SD 24.91), 34.54 dB (SD 25.85), and 50.12 dB (SD 27.27). At most frequencies, there is a statistically significant improvement (Mann–Whitney U Test): 0.5 kHz, *p* < 0.001; 1 kHz, *p* < 0.006; 2 kHz *p* = 0.013; 4 kHz *p* = 0.078 (Figure 1).

The mean hearing threshold for bone conduction before surgery was 13.04 dB (SD 12.64) at 0.5 kHz, 16.04 dB (SD 16.12) at 1 kHz, 25.25 dB (SD 19.40) at 2 kHz, and 26.83 dB (SD 23.33) at 4 kHz, and after the surgery, respectively, 12.38 dB (SD 15.16), 14.96 dB (SD 18.01), 23.42 dB (SD 19.51), and 26.00 dB (SD 22.65). No statistically significant difference was found at any of the frequencies (Mann–Whitney U Test): 0.5 kHz, *p* = 0.322; 1 kHz, *p* = 0.249; 2 kHz, *p* = 0.265; 4 kHz *p* = 0.746 (Figure 2).

### 3.2. STAM

The frequency of cholesteatoma occurrence differed for individual locations, the most often affected being A 83 (69.17%), M 58 (48.33%), and T 55 (45.83%), followed by S1 21 (17.5%) and S2 33 (27.5%).

### 3.3. Stage of Cholesteatoma

The severity of cholesteatoma was assessed by stage according to the AEONO/JOS classification: stage I was found 44 times (36.6%), stage II 62 times (51.7%), stage III 14 times (11.7%), and stage IV was not found in the cohort group. The changes of average bone conduction after surgery between stages of cholesteatoma was statistically significant for stage II at 4 kHz (*p* = 0.048).

### 3.4. SAMEO-ATO

The results of clinical classification of all the cases were retrospectively compared according to the SAMEO classification (Table 1). The type of tympanoplasty performed was recorded according to the ATO classification (Table 2).

### 3.5. Relationship between Cholesteatoma Location and Bone Conduction Frequency

Comparison of bone conduction changes at individual frequencies in relation to cholesteatoma location according to STAM revealed a statistically significant shift in the presence/absence of cholesteatoma in the attic (A) at 4 kHz 1.69 dB (SD 9.92 dB)/−6.49 (SD 10.92) (*p* = 0.0001) (Figure 3), mastoid cavity (M) at 0.5 kHz 0.95 dB (SD 9.29)/−2.18 dB (SD 6.93) (*p* = 0.024); 1 kHz 0.76 dB (SD 9.02)/−2.82 dB (SD 8.76) (*p* = 0.032); 2 kHz 0.09 dB (SD 7.86)/−3.36 dB (SD 9.33) (*p* = 0.039) (Figure 4), and the supratubal recess (S1) at 4 kHz 5.00 dB (SD 9.87)/2.07 (SD 10.71) (*p* = 0.003) (Figure 5). For cholesteatoma in the tympanic cavity (T) and the tympanic sinus (S2), there was no statistically significant difference (Table 3).

The type and extent of surgery (Table 1) according to the SAMEO classification was statistically significant for bone conduction changes in the variables: obliteration of mastoid cavity (O) at 0.5 kHz (*p* = 0.007) and at 1 kHz (*p* = 0.047) (Figure 6); the type of tympanoplasty (Table 2) according to the ATO classification, especially the tympanoplasty type (O), had no statistically significant effect on the bone conduction threshold shift (*p* = 0.128).

## 4. Discussion

It is a well-known fact that cholesteatoma and its surgical treatment influences both air and bone conduction [4]. Cholesteatoma in the middle-ear cavity can cause deterioration in the hearing threshold of bone conduction. There are obviously multiple reasons: the most probable being the effects of products of the cholesteatoma toxic to the membranous labyrinth, and the pressure of granulations, inflammatory exudates, and the cholesteatoma itself on the oval and round windows’ membranes, thus reducing the movement of fluids inside the inner ear, and mechanically influencing stapes movements. The authors assume that cholesteatoma causes deterioration in bone conduction in two ways:Physically, by blocking the movement of the middle-ear bones and oval window membrane, which decreases the motility of liquid and firm structures of the inner ear under sound stimulation. Cholesteatoma impairs the movement of the oval window membrane by direct contact; and contact with middle-ear bones increases impedance and reactance, and hinders movement of the footplate in the oval window. These mechanisms also block the movements of perilymph and thus, indirectly, the movement of hair cells.Chemically, by releasing toxins into inner ear perilymph, which impairs the movement and metabolism of hair cells.

As some of these changes are reversible, removal of the cholesteatoma can lead to restitution of physical and chemical changes. The new condition can then lead to bone conduction improvement. One of the internal factors described is a lower level of microRNA 21 in the inner ear in the occurrence of cholesteatoma. This confirms that cholesteatoma changes the metabolism of the inner ear, and changes the transcription of particular proteins [13].

In the present group, deterioration of bone conduction by more than 5 dB was seen in 14 cases (15 ears), 12.5%, and of them, 5 patients (five ears), 4.7% presented with deterioration of more than 10 dB. A negative prognostic factor is granulations in the tympanic cavity [14]. Other studies, however, did not prove the effect of granulations as a negative prognostic factor [15]. Young people are more liable to decreased bone conduction due to cholesteatoma surgery, especially in revision surgeries [7]. The recurrence of cholesteatoma was always excluded by second look operation or magnetic resonance. These two methods are considered standard procedures in our department.

A transitory decrease in bone conduction in the contralateral ear due to the transfer of the noise of the drill through the cranial bones was also documented [16]. Improvement of the threshold of bone conduction by more than 10 dB, occurring in 2–5% patients, is less often reported [17]. Smaller improvements in bone conduction after chronic otitis media surgery are more common, and in the present cohort, improvement of bone conduction by more than 5 dB was seen in 26 patients (31 ears) (29.5%), and by more than 10 dB in 5 (4.7%) patients, which is in harmony with the literature [17]. Improvement is explained variously by closure of the perilymphatic fistula, unblocking of the membrane of the round window, or removal of cholesteatoma and limitation of action of its toxins. An important positive factor is removal of granulations from the tympanic cavity, and precise preservation of the mucosa in the middle-ear cavity, whereas a negative factor for the threshold of bone conduction proved to be the scars in the round window [9]. Our observations overall logically infer that pre-surgically deteriorated bone conduction shows a greater tendency towards improvement than pre-surgically unchanged bone conduction. This observation contrasts with the general assertion that a pre-surgical decrease in bone conduction is significantly connected with a decrease in bone conduction after surgery [18].

Rosito et al. reported the influence of cholesteatoma size on deterioration of bone conduction in a cohort of patients with cholesteatoma, but did not find any relationship between cholesteatoma location and the degree of deterioration of bone conduction [19]. In the presented cohort, changes of bone conduction after cholesteatoma surgery were compared, and a significant shift of bone conduction was found in patients with chronic inflammation with cholesteatoma in the mastoid cavity (M); if cholesteatoma is localized in the mastoid cavity, there is a greater probability of bone conduction threshold shift after surgery. This theory is supported by the finding that undamaged mucosa of the mastoid cavity is one of the good prognostic factors for post-surgical hearing [20]. Similarly, changes of bone conduction were seen in patients who have cholesteatoma in the attic (A), and in the supratubal recess (S1), versus those who had no cholesteatoma in those locations. A new feature of this study is the statistical evaluation and demonstration of the influence of cholesteatoma location according to STAM classification on the changes of bone conduction after surgery.

The influence of the type of surgery according to SAMEO-ATO classification on improvement of the threshold of bone conduction has not been proven in the present study, except for obliteration of the mastoid cavity (O) which is closely related to cholesteatoma presence in the mastoid cavity (M). Similar statements are found in the literature [7]. The interesting result in our cohort is the statistical significance of obliteration of the mastoid cavity on the bone conduction threshold shift. However, the difference between bone conduction changes, and open and closed mastoidectomies was not proven in previous studies [21].

The otic capsule vibrations, when directly measured, showed correlation with the stimulation frequency, but did not provide a reliable estimate for bone conduction hearing [22]. The direct measurement of intracochlear pressure represents a more precise method, which corresponds more tightly with bone conduction hearing [23]. In our study the influence of mastoid obliteration (O) on bone conduction changes can be explained by the covering of the otic capsule by biomaterials such as bone pate, which influences the vibrations of the otic capsule during stimulation.

The degree of mastoid pneumatization was found important on CT scans for bone conduction auditory threshold shifts, and subjects with small mastoid pneumatization presented a significantly higher hearing loss in bone conduction compared to those in the large mastoid pneumatization group [24]. In our six cases of complete obliteration of the mastoid cavity (O2), there was low mastoid pneumatization; hence, worse results in bone conduction could be expected.

The presence of granulations and exudate in the tympanic cavity in cholesteatoma-free myringoplasties proved to be a negative prognostic factor for bone conduction improvement in patients after surgery [25]. In a group of 181 patients, an intact chain of ossicles had a negative influence on bone conduction [17].

## 5. Conclusions

Middle-ear cholesteatoma classification STAM is applicable to the investigation of the influence of cholesteatoma location on post-surgical changes of hearing threshold for bone conduction. The changes are frequency-specific for individual locations of cholesteatoma. For changes of the hearing threshold for bone conduction at 0.5, 1, and 2 kHz, the occurrence of cholesteatoma in the mastoid cavity (M) is of importance; the presence of cholesteatoma in the attic (A) and supratubal recess (S1) is a statistically significant factor for changes at 4 kHz; for other locations of cholesteatoma (S2, M), no influence on changes of bone conduction was found. Cholesteatoma surgery according to the SAMEO-ATO classification was statistically significant in the parameter: obliteration of mastoid cavity (O) at 0.5 kHz and 1 kHz.

## Figures and Tables

**Figure 1 jcm-11-04481-f001:**
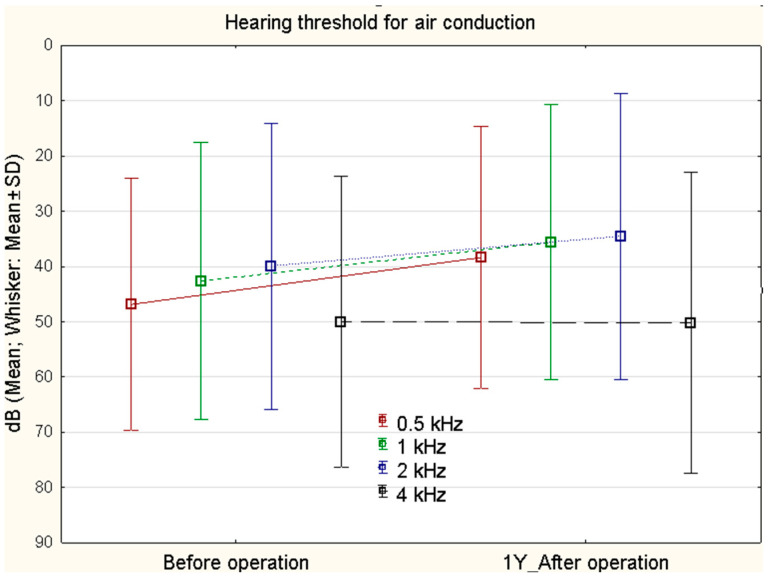
Air conduction (0.5, 1, 2, 4 kHz) before surgery and 1 year after surgery (*n* = 120). At most frequencies, there is a statistically significant improvement. Mann–Whitney U Test: 0.5 kHz (*p* < 0.001), 1 kHz (*p* < 0.006), 2 kHz (*p* = 0.013) and 4 kHz (*p* = 0.078).

**Figure 2 jcm-11-04481-f002:**
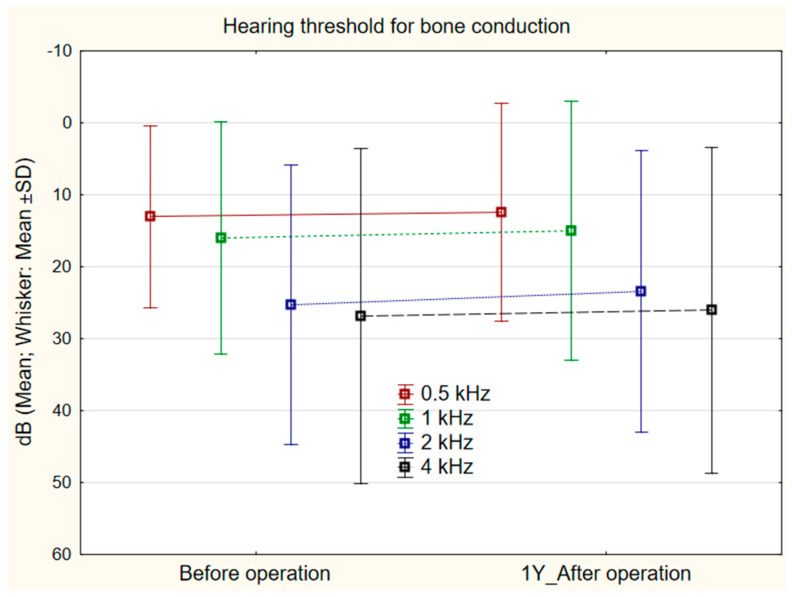
Hearing threshold for bone conduction (0.5, 1, 2, 4 kHz) before and 1 year after surgery (*n* = 120). No statistically significant difference was found at any frequency. Mann–Whitney U Test: 0.5 kHz (*p* = 0.322), 1 kHz (*p* = 0.249), 2 kHz (*p* = 0.265), 4 kHz (*p* = 0.746).

**Figure 3 jcm-11-04481-f003:**
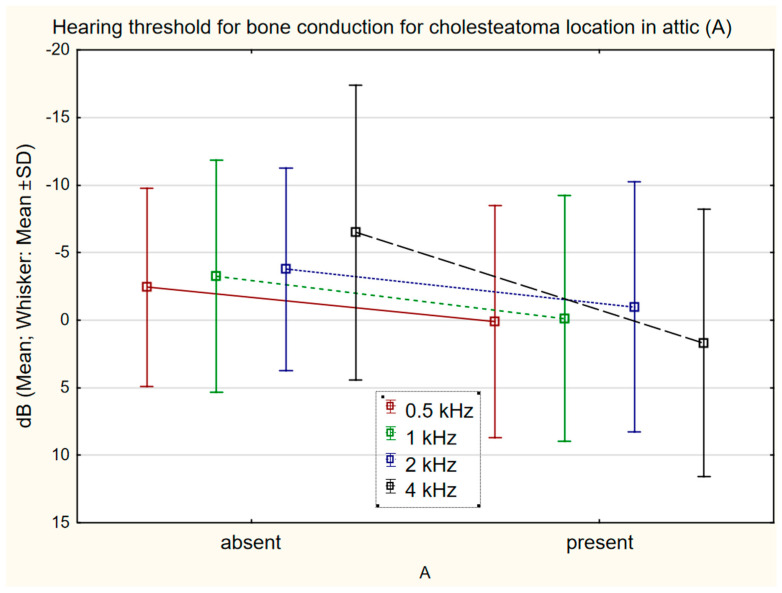
The difference in bone conduction prior to surgery and 1 year after surgery in presence (*n* = 83) or absence (*n* = 37) of cholesteatoma in the attic (A) at 4 kHz 1.69 dB (SD 9.92)/−6.49 dB (SD 10.92) (*p* = 0.0001).

**Figure 4 jcm-11-04481-f004:**
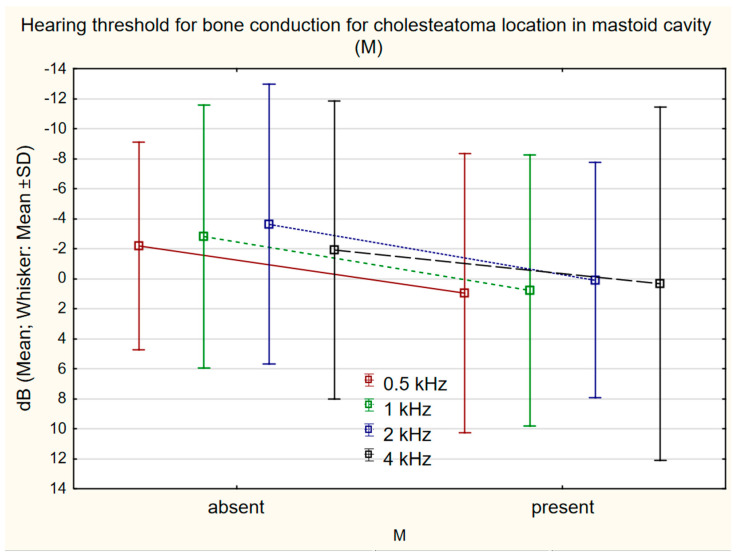
The difference in bone conduction prior to surgery and 1 year after surgery in presence (*n* = 58) or absence (*n* = 62) of cholesteatoma in the mastoid cavity (M) at 0.5 kHz 0.95 dB (SD 9.29)/−2.18 dB (SD 6.93) (*p* = 0.024), 1 kHz 0.76 dB (SD 9.02)/−2.82 dB (SD 8.76) (*p* = 0.032), 2 kHz 0.09 (SD 7.86)/−3.36 (SD 9.33) (*p* = 0.039).

**Figure 5 jcm-11-04481-f005:**
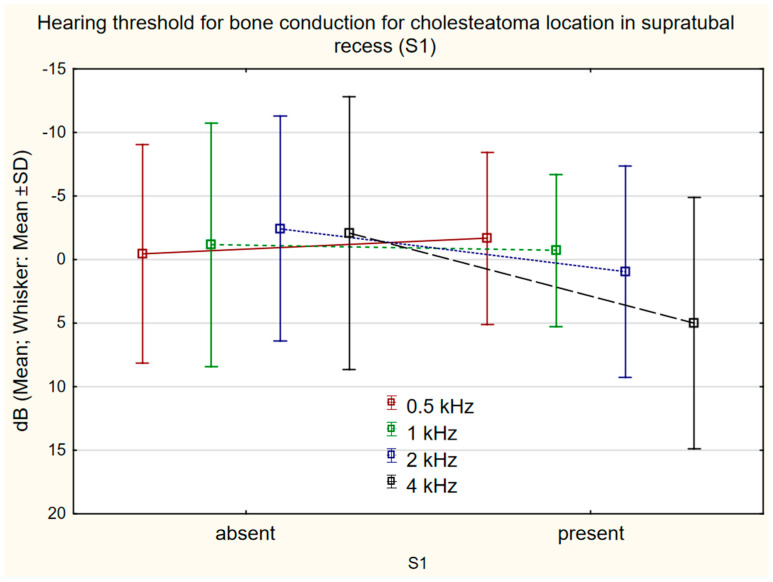
The difference in bone conduction prior to surgery and 1 year after surgery in presence (*n* = 21) or absence (*n* = 99) of cholesteatoma in the supratubal recess (S1) 4 kHz 5.00 dB (SD 9.87)/2.07 (SD 10.71) *p* = 0.003).

**Figure 6 jcm-11-04481-f006:**
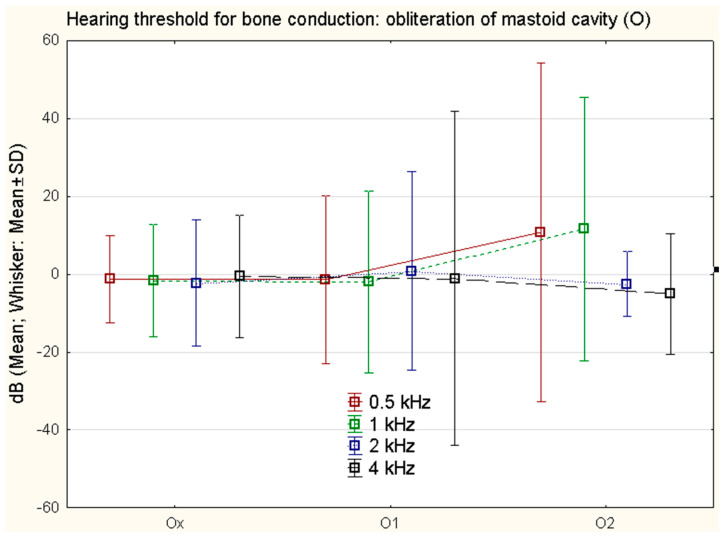
The difference in bone conduction prior to surgery and 1 year after surgery in different types of mastoid cavity obliteration. Ox no reconstruction, O1 partial obliteration, O2 total obliteration; at 0.5 kHz (*p* = 0.007) and at 1 kHz (*p* = 0.047).

**Table 1 jcm-11-04481-t001:** Occurrence of individual types of surgical treatment of chronic otitis media with cholesteatoma in the group of 92 patients (120 ears), according to the SAMEO classification.

S		A		M		E		O	
S1	120	A3	28	Mx	21	Ex	48	Ox	96
		A4	92	M1a	4	E1	19	O1	18
				M1a + 2a	46	E2	53	O2	6
				M2a	18				
				M2b	6				
				M2c	20				
				M3a	4				
				M3b	1				

**Table 2 jcm-11-04481-t002:** Occurrence of individual types of tympanoplasty in the group of 92 patients (120 ears) according to the ATO classification.

A		T		O	
Ax	115	Tx	12	Ost	14
A1	4	T1	4	On	13
A2	1	T2	102	Ox	15
		T3	2	Osd	52
				Oft	18
				Ofd	8

**Table 3 jcm-11-04481-t003:** The relationship between cholesteatoma location according to STAM and the frequency of bone conduction shifts. Test ANOVA, *n* = 120. For the occurrence of cholesteatoma in the attic (A), a statistically significant difference at 4 kHz (*p* < 0.001) was found, in the supratubal recess (S1) at 4 kHz (*p* = 0.003), and for the mastoid (M) at 0.5 kHz (*p* = 0.024), at 1 kHz (*p* = 0.032), and 2 kHz (*p* = 0.039).

Location/ *p*-Value	0.5 kHz	1 kHz	2 kHz	4 kHz
**A**	0.087	0.053	0.189	**0.0001**
**T**	0.497	0.681	0.745	0.742
**M**	**0.024**	**0.032**	**0.039**	0.476
**S1**	0.683	0.712	0.103	**0.003**
**S2**	0.684	0.841	0.473	0.555

## Data Availability

Data can be obtained from the authors upon reasonable request.

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
