# Peer review of "The Relationship between Bone Conduction Hearing Threshold Shifts after Surgery for Chronic Otitis Media with Cholesteatoma According to STAM, EAONO/JOS, and SAMEO-ATO Classifications"

_jcm, 2022, doi:10.3390/jcm11154481_

Round 1
Reviewer 1 Report
The manuscript has been improved. However, I have some other comments:
1. Need further English editing
2. As the staging was based on the number of sublocations, it is interesting to know in a more details of the combination of the sublocations that causing the problem of bone conduction at 4kHz
3. For the SAMEO-ATO classification, to describe the meaning of the number next to it in the methodology. For example: A3, A4 (3 or 4 means what?)
4. Table 3 has sufficiently described graphs 3,4 and 5
5. What is meant by “statistical different in parameter” in the last paragraph in the results?
Author Response
Thank you for your constructive approach. Suggested changes were included in the text.
- Need further English editing
Answer: Further English editing done.
- As the staging was based on the number of sublocations, it is interesting to know in a more details of the combination of the sublocations that causing the problem of bone conduction at 4kHz
Answer: Thank you for this valuable comment. It is really interesting. We performed several different calculations for different combinations, some of them seemed to be significant. However, this approach does not correspond with established STAM classification, the classification knows only cholesteatoma sublocations and stage of cholesteatoma and does not reflect different combinations. As the paper is based on these classifications, we would like to respect the original concept with its potential disadvantages. Nevertheless, we will investigate this issue in our further work.
- For the SAMEO-ATO classification, to describe the meaning of the number next to it in the methodology. For example: A3, A4 (3 or 4 means what?)
Answer: Added.
For mastoid surgery
S1 primary surgery, S2p – planned 2nd surgery, S2r – revision surgery
A 1 Endoscopic 2 Transcanal 3 Endaural, 4 Retrouaricular
Mx - no mastoidectomy, M1a canal wall preserved, M1b canal wall preserved and posterior tympanotomy, M2a only scutum removed, M2b scutum removed and postero-superior canal wall removed, M2c whole canal removed, M3a subtotal petrosectomy with preservation of the otic capsula, M3b subtotal petrosectomy with removal of the otic capsula, including labyrinthectomy.
E1 no reconstruction, E2 soft material reconstruction, E2 hard material reconstruction
Ox no reconstruction, O1 partial reconstruction, O2 total reconstruction
For middle ear surgery
Ax No bone canal removal, A1 tympanic sulcus widening A2 external canal widening A3 total canalplasty
Tx no ear drum grafting, Tn Original ear drum preserved T1 intact membrane reinforcement, T2 partial grafting, T3 Total ear drum grafting
Ost tympanic membrane to stapes head, On intact chain preservation, Ox no reconstruction performed, Osd Ear drum directly repositioned to stapes head, Oft reconstruction between ear drum and foot plate of the stapes, Ofd ear drum directly repositioned onto stapes footplate.
- Table 3 has sufficiently described graphs 3, 4 and 5
Answer: It is true, that some information in the graphs and tables are doubled, but the graphs shows comprehensively the hearing threshold levels and bone conduction changes after the surgery and Table 3 highlights the statistical power of the values showed in graphs. In the previous revisions we showed all data in one graph, but then was requested by reviewers to simplify graphs and statistical outcomes.
- What is meant by “statistical different in parameter” in the last paragraph in the results?
Answer: Corrected:
The type and extent of surgery (Table 1) according to the SAMEO classification was statistically significant for bone conduction changes in variable: obliteration of mastoid cavity (O) at 0.5 kHz (p=0.070) and at 1 kHz (p=0.047); the type of tympanoplasty (Table 2) according to the ATO classification, especially the type of tympanoplasty (O) had no statistically significant effect on bone conduction threshold shift (p=0.128).
Reviewer 2 Report
The relationship between bone conduction hearing threshold shifts after surgery for chronic otitis media with cholesteatoma according to STAM and SAMEO-ATO classifications
Review 12-17.06.2022 ID 2925279 Journal of Clinical Medicine
ID jcm-1770606
This study aimed to find whether changes in bone conduction after cholesteatoma surgery are bound to a specific location and whether the possible dependence is frequency specific.
The problem of sensorineural hearing loss as a complication of chronic ear surgery has been reported in the literature for several years. The mechanism of origin of the hearing loss is explained, among other things, by the presence of a labyrinthine fistula, the manipulation of the surgeon in the area of the oval and round window and the biochemical influence of chronic cholesteatoma otitis media on the inner ear.
In the present study, the authors revisit the topic of bone conduction hearing loss associated with surgery for chronic otitis media, using data on the location of cholesteatoma according to STAM or the type of surgical treatment of chronic otitis media according to SAMEO-ATO.
Although not new, the topic is clinically significant in planning surgical procedures and predicting complications of hearing threshold deterioration so that it can be discussed with the patient before cholesteatoma surgery.
However, there are some comments.
In the Introduction, the authors should refer to more recent literature items (the Introduction includes a citation from 1984) and explain the complex mechanism of bone conduction (through the cranial bones and the capsula otica).
In lines 28-32, the authors described two paths of hearing impairment: chemical (metabolic) damage to the hair cells and mechanical blockage of the round and oval windows. The explanation of how ossicular chain reconstruction or tympanoplasty procedures can affect bone conduction should be moved to the Introduction and need a citation (paragraphs 181 to 188).
In the Materials and Methods
Data from the medical history of patients are missing. For example, did they have a severe hearing loss in the opposite ear? How long before surgery did the patients suffer from chronic otitis with cholesteatoma? Were there other factors that could potentially damage hearing during the one-year follow-up period ( e.g. chemotherapy)? In summarizing, what the inclusion and exclusion criteria of the study group are?
Was CT imaging performed, and was the extent of cholesteatoma performed on this basis, or was it performed during surgery?
The authors wrote that the opposite ear was masked during the audiometry test. Has the ear canal remained closed or open? Please complete the test description? Have any other audiological tests been performed, e.g. bone ABR or otoacoustic emission, if possible? If so, can they be added to additional materials? The audiometric test remains a subjective test, and following the cross-check principle; it is worth having confirmation in other tests.
In the Result section, the authors present graphs of changes in bone conduction before and after surgery. Still, there is no table or chart on the relationship between the type of surgery/ tympanoplasty and hearing threshold. In my opinion, adding a description of the hearing would be worthwhile after CWU and CWD surgery, considering reconstructions and obliteration. It could be fascinating how reconstructing anatomical conditions or obliteration affects bone conduction hearing.
In the Discussion
Paragraphs 203-205 should be moved to the Material and Method section.
The discussion should be improved and supplemented with information on bone conduction pathways, explaining the relationship between the location of the cholesteatoma and the possible effect on bone conduction.
The discussion should be enriched with more citations, also concerning sensorineural hearing loss associated with the surgery of cholesteatoma and what should be compared with the results obtained in work.
The sentence235-236 is unclear:" The influence of the type of surgery according to SAMEO ATO classification on the improvement of a threshold of bone conduction has not been proven in the present study". However, the results in lines 155-158 confirm the statistical relationship.
Conclusions should also refer to the classification of the SAMEO ATO classification since this was the purpose of the work.
Author Response
Thank you for your constructive approach. Suggested changes were included in the text. This is the compilation of changes.
- In the Introduction, the authors should refer to more recent literature items (the Introduction includes a citation from 1984) and explain the complex mechanism of bone conduction (through the cranial bones and the capsula otica).
Answer: Citation added.
- In lines 28-32, the authors described two paths of hearing impairment: chemical (metabolic) damage to the hair cells and mechanical blockage of the round and oval windows. The explanation of how ossicular chain reconstruction or tympanoplasty procedures can affect bone conduction should be moved to the Introduction and need a citation (paragraphs 181 to 188).
Answer: Paragraphs moved to introduction
- Due to its locally destructive potential, classifications such as those of tumor processes have even been proposed [9]. Classification based on the size of cholesteatoma, condition of the ossicles, and the presence of complications aimed to set up an ascending scale which would correspond to clinical importance [10]. Every change in cholesteatoma classification is a challenge for re-examination of complications, hearing gains, or surgical procedures [10,11]. This report evaluates newly originated cholesteatoma STAM and SAMEO-ATO classifications in relation to post-surgically improved bone conduction.
- In the Materials and Methods Data from the medical history of patients are missing. For example, did they have a severe hearing loss in the opposite ear? How long before surgery did the patients suffer from chronic otitis with cholesteatoma?
Answer: Those data are not in our data file and cannot be evaluated.
- Were there other factors that could potentially damage hearing during the one-year follow-up period (e.g. chemotherapy)? In summarizing, what the inclusion and exclusion criteria of the study group are?
Answer: None of included patients underwent chemotherapy treatment during follow-up period nor were exposed to other factors that could potentially damage hearing. We will evaluate these factors for other forthcoming study on recidivism of cholesteatoma.
As for the versatility of used classifications we were able to include all patients who underwent surgery in our department. Exclusion criterion for our study were missing audiology data.
- Was CT imaging performed, and was the extent of cholesteatoma performed on this basis, or was it performed during surgery?
Answer A: Added:
High Resolution Computer Tomography was always used before surgery to evaluate temporal bone erosions.
Answer B: added:
Classification of cholesteatoma according to STAM was performed retrospectively, from patient’s files, mainly from surgical protocols.
- The authors wrote that the opposite ear was masked during the audiometry test. Has the ear canal remained closed or open? Please complete the test description? Have any other audiological tests been performed, e.g. bone ABR or otoacoustic emission, if possible? If so, can they be added to additional materials? The audiometric test remains a subjective test, and following the cross-check principle; it is worth having confirmation in other tests.
Answer: corrected:
Pure tone audiometry was performed on affected and non-affected ear at frequencies of 0.5, 1, 2, and 4 kHz, to find the hearing threshold of air and bone conduction, prior to surgery and 1 year after surgery. In During the examination, the contra-lateral ear was masked, with respect of audiological masking protocols. The examinations were done repeatedly to avoid bias. The tympanometry and speech audiometry had been done before surgery, if applicable. However, it was not the matter of publication.
- In the Result section, the authors present graphs of changes in bone conduction before and after surgery. Still, there is no table or chart on the relationship between the type of surgery/ tympanoplasty and hearing threshold. In my opinion, adding a description of the hearing would be worthwhile after CWU and CWD surgery, considering reconstructions and obliteration. It could be fascinating how reconstructing anatomical conditions or obliteration affects bone conduction hearing.
Answer A: Added paragraph in results. As the SAMEO-ATO doesn`t include categories CWU and CWD, we used corresponding M (mastoid surgery) classification as well as other parts of SAMEO-ATO for evaluating effect on bone hearing conduction.
Answer B: This result was rather questionable, but I added it in the results.
- In the Discussion Paragraphs 203-205 should be moved to the Material and Method section.
Answer: Moved.
- The discussion should be improved and supplemented with information on bone conduction pathways, explaining the relationship between the location of the cholesteatoma and the possible effect on bone conduction.
Answer: added:
The otic capsule vibrations, when directly measured, showed correlation with stimulation frequency, but does not provide a reliable estimate in bone conduction hearing [22]. The direct measurement of intracochlear pressure represents more pre-cise method, which corresponds more tightly with bone conduction hearing[23]. In our study the influence of mastoid obliteratin (O) to bone conduction changes can be ex-plained by covering the otic capsule by biomaterials as is the bone pate, which influ-ences the vibrations of otic capsule during stimulation.
The Size of mastoid pneumatization was found important on CT scans for bone conduction auditory threshold shifts, and the persons with small mastoid pneumatiza-tion revealed significantly higher hearing loss in bone conduction compared to large mastoid pneumatization group[24]. In our 6 cases of complete obliteration of mastoid cavity (O2), was small mastoid pneumatization and thus worse results in bone conduc-tion can be expected.
- The discussion should be enriched with more citations, also concerning sensorineural hearing loss associated with the surgery of cholesteatoma and what should be compared with the results obtained in work.
Answer: added in introduction section.
Several factors exert an influence on the post-surgical hearing status. Serious de-terioration of hearing threshold of bone conduction, which Tos describes in ca 1.2% of cases, is influenced by acoustic trauma caused by the drilling, perilymphatic fistula, and undesirable manipulations on middle-ear ossicles [5]. The manipulation with ossi-cles can cause temporary auditory threshold shift on 2 and 4 kHz, but the drilling in mastoid can cause permanent auditory threshold shift at 4 kHz [10].
However, the difference between bone conduction changes and opened and closed mastoidectomies was not proven [21].
- The sentence 235-236 is unclear:" The influence of the type of surgery according to SAMEO ATO classification on the improvement of a threshold of bone conduction has not been proven in the present study". However, the results in lines 155-158 confirm the statistical relationship.
Answer: Corrected
Cholesteatoma surgery according to the SAMEO-ATO classification was statistically significant in parameter: obliteration of mastoid cavity (O) for 0.5kHz (p=0.007) and 1 kHz (p=0.047).
- Conclusions should also refer to the classification of the SAMEO ATO classification since this was the purpose of the work.
Answer: Added.
Cholesteatoma surgery according to the SAMEO-ATO classification was statistically significant in parameter: obliteration of mastoid cavity (O) for 0.5kHz (p=0.007) and 1 kHz (p=0.047).
Round 2
Reviewer 1 Report
The manuscript has been sufficiently improved.
Author Response
Thank you for your kind review. English language was corrected by native speaker specialized in medicine.
Reviewer 2 Report
Dear authors,
Thank you for the corrections made to the paper. Please include the inclusion and exclusion criteria of the study group in the description of participants in the Materials and Methods section.
Author Response
Thank you for your kind review.
This text was added:
Inclusion criteria: Chronic otitis media with cholesteatoma (histologically proven), primary surgery, ability of repeated audiological testing, one year follow up
Exclusion criteria: Revision surgery, recidivation of cholesteatoma, malignancies in follow up period, deaf ear, inability of audiological testing, missing audiological data, operation less than one year before.
This manuscript is a resubmission of an earlier submission. The following is a list of the peer review reports and author responses from that submission.
Round 1
Reviewer 1 Report
This paper describes the postoperative improvement of bone conduction of the cholesteatoma location.
What is the main mechanism of the bone conduction change according to the location of the cholesteatoma ?
There should be additional description on the audiological and middle ear mechanical reasons in the discussion section.
This is essential for readers.
Author Response
Thank you for your kind advices. Your suggestions have been added to the text.
Answer: The authors assume that cholesteatoma deteriorates bone conduction in two ways:
- physically by blocking the movement of the middle ear bones and oval window membrane, which decreases the motility of liquid and firm structures of the inner ear while sound stimulation. Cholesteatoma impairs the movement of oval window membrane by direct contact, the contact with middle ear bones increases impedance and reactance and hinders movement of the footplate in the oval window. These mechanisms also block the movements of perilymph and thus, indirectly, the movement of hair cells.
- chemically by releasing toxins into the inner ear perilymph, which impairs the movement and metabolism of hair cells.
As some of these changes are reversible, removal of the cholesteatoma can lead to restitution of physical and chemical changes. The new condition can than lead to bone conduction improvement.
Reviewer 2 Report
This study evaluates the improvement of bone conduction thresholds after surgical resection of cholesteatoma. The location and classification of cholesteatoma are evaluated in their effects on post-surgical bone conduction. This article has several major methodologic flaws. First, the recurrence/incomplete resection of cholesteatoma is common and the most common reason for changes in hearing thresholds 1 year out. The authors test the hearing at 1 year post-surgery but do not control for whether cholesteatoma has recurred at this time, an obvious influence on post-operative hearing thresholds. This would typically be done with either a second look procedure to confirm the absence of cholesteatoma or an MRI.
Furthermore, an improvement of bone conduction after the removal of cholesteatoma is expected with a good surgical approach including full removal of cholesteatoma and ossicular chain reconstruction/tympanic membrane repair.
The location of cholesteatoma having an influence on bone conduction is not a novel finding. For example, choleasteatoma involving the ossicular chain should have a higher influence on bone conduction than choleseteatoma not involving the ossicular chain. Although this article uses a standard classification system, this information still fails to add a significant contribution to the literature.
Lastly, the statistical approach is flawed. The authors artificially created two groups based on whether or not they meet a certain threshold of hearing change post-operatively. They then run statistical analysis on these two artificially created groups. Therefore, the clinical significance of this statistical change is unknown.
Author Response
Thank you for your kind advices. Your suggestions have been added to the text.
This study evaluates the improvement of bone conduction thresholds after surgical resection of cholesteatoma. The location and classification of cholesteatoma are evaluated in their effects on post-surgical bone conduction. This article has several major methodologic flaws. First, the recurrence/incomplete resection of cholesteatoma is common and the most common reason for changes in hearing thresholds 1 year out. The authors test the hearing at 1 year post-surgery but do not control for whether cholesteatoma has recurred at this time, an obvious influence on post-operative hearing thresholds. This would typically be done with either a second look procedure to confirm the absence of cholesteatoma or an MRI. 
Answer: The recurrence of cholesteatoma could be always excluded by second look operation or magnetic resonance. These two methods are considered as standard procedures in our department.
Furthermore, an improvement of bone conduction after the removal of cholesteatoma is expected with a good surgical approach including full removal of cholesteatoma and ossicular chain reconstruction/tympanic membrane repair. 
Answer: Removal of cholesteatoma and ossicular chain reconstruction are basic requirements for bone conduction improvement. The authors try to identify if tympanoplasties can affect bone conduction as well.
The location of cholesteatoma having an influence on bone conduction is not a novel finding. For example, choleasteatoma involving the ossicular chain should have a higher influence on bone conduction than choleseteatoma not involving the ossicular chain. Although this article uses a standard classification system, this information still fails to add a significant contribution to the literature. 
Answer: Even though the fact that cholesteatoma affects bone conduction is not new, authors try to support this information by real data and statistical analysis.
Lastly, the statistical approach is flawed. The authors artificially created two groups based on whether or not they meet a certain threshold of hearing change post-operatively. They then run statistical analysis on these two artificially created groups. Therefore, the clinical significance of this statistical change is unknown. 
Answer: Cut off criteria for this kind of statistics had to be set, mainly to determine which values can be considered as bone conduction improvement. The criteria were set according to the data from the literature. This is a standard procedure in a homologous group which has to be devided into two different parts, so that the two parts are statistically different. The factor, that differentiates these two groups can then be found. In our case we study the SAMEO ATO classification.
Reviewer 3 Report
The authors studied the relationship between the bone conduction and the STAM and SAMEO ATO classifications in post surgery cases of cholesteatoma. It was an interesting study as the finding may be used for the discussion of the prognosis of hearing loss in a case of cholesteatoma.
Comments:
- The word "unexpected" in the title is not needed
- The exclusion of those having deterioration in bone conduction may gives a false conclusion
- The symbols used in the graphs are not clear
- It is not clear whether the graphs are showing the differences between pre and post surgery or between group 1 and 2 for each of the frequencies tested
- Table 1 and 2 need to be improved for better understanding. The abbreviations used need to be spelled out
Author Response
Thank you for your kind advices. Your suggestions have been added to the text.
The authors studied the relationship between the bone conduction and the STAM and SAMEO ATO classifications in post surgery cases of cholesteatoma. It was an interesting study as the finding may be used for the discussion of the prognosis of hearing loss in a case of cholesteatoma.
Comments:
- The word "unexpected" in the title is not needed
Answer: deleted
- The exclusion of those having deterioration in bone conduction may gives a false conclusion
Cut off criteria for this kind of statistics had to be set, mainly to determine which values can be considered as bone conduction improvement. The criteria were set according to the data from the literature. This is a standard procedure in a homologous group which has to be devided into two different parts, so that the two parts are statistically different. The factor, that differentiates these two groups can then be found. In our case we study the SAMEO ATO classification.
- The symbols used in the graphs are not clear
Answer: corrected, simplified and visualisation improved.
- It is not clear whether the graphs are showing the differences between pre and post surgery or between group 1 and 2 for each of the frequencies tested
Answer: Corrected, extra heading and titles added.
- Table 1 and 2 need to be improved for better understanding. The abbreviations used need to be spelled out.
Answer: the tables were changed, and the highlights corrected for better understanding.
Round 2
Reviewer 3 Report
The manuscript has been improved.
Further comments:
- The definition of group 1 and 2 need to be stated in the abstract
- The authors need to explain about the significant differences in bone conduction between the two groups prior to surgery. The significant differences may affect the study as the research was done to see the improvement in hearing by comparing both of the groups
- I believed the text need further English editing
Author Response
Thank you for your constructive approach. I corrected the text in respect of your valuable advices.
- The definition of group 1 and 2 need to be stated in the abstract
Answer: sentence in abstract added to define the Groups.
Group 1 (26 patients with 31 ears) improved BC by more than 5 dB, Group 2 (52 patients with 74 ears) by less than 5dB .
- The authors need to explain about the significant differences in bone conduction between the two groups prior to surgery. The significant differences may affect the study as the research was done to see the improvement in hearing by comparing both of the groups
Answer:The cohort was divided arteficially in two unequal subgroups, deploying cutt of criteria based on mean difference of bone conduction hearing threshold shifts before and after surgery. After that we analysed the location of cholesteatoma but not the hearing improvement. The rewiwer is right it would mistakenly influence our results, but we did not do that.¨
To elucidate this unclear, weak point we added this text:
The cohort was divided artificially in two unequal subgroups, deploying cut of cri-teria based on mean difference of bone conduction hearing threshold shifts before and after surgery. After that we analyzed the location of cholesteatoma, during surgery in both groups. This method must not be used to compare the bone conduction hearing thresholds, it would affect the results.
- I believed the text need further English editing
Answer: the text was edited further, as required.